# A 5-Year-Old Child with a Deep Neck Abscess Complicated by Laryngeal Obstruction

**DOI:** 10.3390/children10010017

**Published:** 2022-12-22

**Authors:** Armando Di Ludovico, Massimiliano Raso, Paola Di Filippo, Sabrina Di Pillo, Roberta Zappacosta, Giustino Parruti, Pasquale Zingariello, Francesco Chiarelli, Marina Attanasi

**Affiliations:** 1Department of Pediatrics, University of Chieti, 66100 Chieti, Italy; 2Department of Pathology, University of Chieti, 66100 Chieti, Italy; 3Infectious Diseases Unit, Pescara General Hospital, 65123 Pescara, Italy; 4Department of Medical Oral Sciences, and Biotechnology, Section of Otolaryngology, University of Chieti, 66100 Chieti, Italy

**Keywords:** torticollis, neck pain, deep neck space infection, neck abscess

## Abstract

Deep neck space infections (DNSI) are defined as infections in the potential spaces and fascial planes of the neck. We show the clinical case of a retro and para-pharyngeal abscess in a healthy 5-year-old child complicated by compression and dislocation of the larynx with marked airway caliber reduction and potentially fatal extension up to the mediastinal aditus. DNSI can occur at any age and, due to its rapid progression, requires immediate treatment in children. In healthy children, concurrent abscesses in separate neck spaces are rare. DNSI recurrence should alert the physician to the possibility of a congenital problem, and if imaging fails, laryngoscopy may be the best diagnostic technique.

## 1. Introduction

Deep neck space infections (DNSI) are defined as infections in the potential spaces and fascial planes of the neck. DNSI can occur at any age and requires prompt management in the pediatric age group because of its rapidly progressive nature. Concurrent abscess in the distinct neck spaces has rarely been reported in healthy children. Hence, we show the clinical case of a retro and para-pharyngeal abscess in an otherwise healthy 5-year-old child complicated by compression and dislocation of the larynx. Additionally, the abscess induced a marked airway caliber reduction and potentially fatal extension up to the mediastinal aditus. Additional exams were performed for a potential underlying congenital abnormality as the recurrence of DNSI in the patient’s past medical history.

## 2. Case Report

### 2.1. Acute Presentation

We describe the case of a 5-year-old boy admitted to our Pediatric Department, in Chieti (Italy), due to fever and left cervical region neck pain, dysphonia, dysphagia, and odynophagia. Symptoms occurred six days prior to hospitalization. The patient was firstly treated at home without any clinical improvement with amoxicillin/clavulanate (60 mg/kg/day) for three days, then with azithromycin (10 mg/kg/day) for another three days, and with betamethasone (0.1 mg/kg/day) for one day. In his past medical history, two similar episodes, characterized by fever and left neck pain, were described: the first one at age 3 years and treated with amoxicillin/clavulanate (50 mg/kg/day) for five days with clinical improvement; the second one at 6 months before this hospitalization and was self-limited. There was no medical history of immunosuppression. There was no other medical condition or physical abnormality to be considered. Upon admission, the child showed uncomfortable conditions characterized by fever (38.2 °C), cervical pain, and restricted cervical range of motion (Table 1: differential diagnosis in children with neck pain and torticollis). Neck swelling and abnormal head posture were not evident. Medical examination showed pharyngeal hyperemia and bilateral cervical and submandibular lymphadenopathy without neck stiffness. Vital signs were normal (heart rate: 138 beats per minute; oxygen saturation: 100% breathing room air; respiration rate: 22 breaths per minute; systolic blood pressure: 96 mmHg; diastolic blood pressure: 68 mmHg). Cardiorespiratory examination was normal. Blood exams showed neutrophilic leukocytosis (white blood cells 16,040/μL, neutrophils 12,410/μL, lymphocites 2180/μL, monocytes 1390/μL), and increased C-Reactive Protein (128.40 mg/L; normal value < 10 mg/L), but normal procalcitonin value (0.120 ng/mL; normal value: less than 0.5 ng/mL). Hemoglobin value was 13 g/dL (normal value: 11.4–14.3 g/dL). Severe acute respiratory syndrome coronavirus 2019 (SARS-CoV-2) real-time-PCR nasopharyngeal swab was negative. After the first clinical evaluation, to rule out a neck abscess, we decided to start an empirical broad-spectrum antibiotic treatment with intravenous ceftriaxone (90 mg/kg/day).

We performed neck and brain magnetic resonance (MRI) as there was no improvement of general malaise and neck pain after 24 h of treatment. Neck MRI showed the presence of a voluminous expansive process (32 × 25 × 85 mm latero-lateral ×antero-posterior ×cranio-caudal) in the retro and para-pharyngeal spaces with extension up to mediastinal aditus (Figure 1A). In addition, the exam showed a significant compression and dislocation to the right of the larynx (Figure 1B), with marked caliber reduction (maximum diameter of 3 mm at the level of the glottic plane). Neck MRI also showed diffuse edema imbibition, and contrast impregnation of the lateral and deep muscles, and fascial planes of the bilateral cervical region (more evident on the left) (Figure 1C), including the prevertebral fascia, the longus capitis muscle, the “danger space” (Figure 1D), and the left thyroid lobe (Figure 1E). Brain MRI was normal. Considering the MRI findings and worsening of the clinical manifestations after 36 h of intravenous antibiotic, the patient underwent the drainage of the lesion by open neck surgery. After surgery, a histological examination was carried out. Microscopic exam showed the involvement of connective muscle and thyroid cells in the suppurative process (Figure 2). No cytologic atypia was detected (Figure 2). We also investigated the thyroid function by blood exam (i.e., thyrotropin, thyroxine, anti-thyroglobulin, anti-thyroid peroxidase antibodies), considering the involvement of thyroid tissue in the histological sample. All those exams were normal. All cultures and additional testing performed were negative (Table 2). The patient continued the treatment with broad-spectrum antibiotics due to the involvement of the “danger space” with mediastinal extension and a high risk of developing other severe complications, although the cultural results were negative. Subsequently, ceftriaxone was replaced with meropenem (at dosage of 55 mg/kg/day ter in die—TID) and clindamycin (at dosage of 36 mg/kg/day TID) in order to provide optimal coverage for Gram-negative bacilli, anaerobes, and potentially resistant Gram-positive cocci (i.e., *Methicillin-Resistant Staphylococcus aureus* (MRSA)). After evaluating past medical history, clinical, and radiological findings, the presence of a deep neck space infection was considered.

The echocardiogram and chest X-ray (CXR) performed during hospitalization were normal. Neck MRI after surgery showed the reduction in retro and left parapharyngeal inflammatory collection, with residual edema imbibition of the left lateral cervical soft tissues (maximum size of 30 × 21 × 55 mm) between the left thyroid space and sternocleidomastoid muscle (Figure 1F). Airway dislocation was also reduced (Figure 1F).

Six days after the abscess drainage, the child showed an important clinical improvement, with a resolution of neck pain and restored cervical mobility. After 72 h of apyrexia, intravenous administration of broad-spectrum antibiotics was discontinued (a total of 10.5 days), and oral therapy with clindamycin (at dosage of 20 mg/kg/day TID) was administered for 10.5 days. After 21 days, the patient was discharged in good general health conditions with clinical, trans-nasal endoscopy and imaging follow-up program.

### 2.2. Investigations for Recurrent Neck Abscesses

One month after discharge, a third neck MRI was performed, showing the absence of para-laryngeal-tracheal fluid collections (Figure 3 and Figure 4) and a small cyst of the right thyroid lobe of 7 mm with homogeneous fluid content. Considering the lesion localization and thyroid gland involvement, branchial arch anomalies were investigated with a laryngoscopy. Laryngoscopy showed the suspicion of a pyriform sinus fistula (PSF). Firstly, the diagnosis of PSF was not confirmed for the presence of residual inflammatory fluid collection. One month later, a second laryngoscopy was performed and the diagnosis of PSF was confirmed (Figure 5: case report timeline). Afterward, we lost the patient because the child was referred to a Pediatric Surgery Center.

## 3. Discussion

Deep neck space infections are defined as infections in the potential spaces and fascial planes of the neck. Clinical diagnosis of DNSI in children might be difficult. Typical and atypical symptoms should be promptly recognized to define correct management and prevent severe complications. Fever, torticollis, neck pain, and odynophagia are the most typical symptoms associated with DNSI [1]. Atypical signs and symptoms can include dysphagia, sialorrhea, dysphonia, dyspnea, thoracic pain, and tongue base pain.

Our patient presented fever, neck pain, limitation of neck movements, and odynophagia as typical symptoms for DNSI, and dysphagia and dysphonia as atypical ones. Neck swelling, abnormal head posture, and neck stiffness were not evident. The diagnostic process of DNSI consists of the evaluation of clinical symptoms and signs, and imaging tests. In order to define accurate management, imaging investigation is pivotal to establish the exact anatomical extent and the potential associated complications [2].

Ultrasound (US) imaging is commonly the first used radiological imaging method when a neck infection is suspected in children. US is quick, non-invasive, free of radiation, and can be performed immediately in the emergency department [3]. It is useful to detect superficial inflammation and percutaneous drainage of superficial abscesses. However, US has low sensitivity and specificity for detecting DNSI and has a poor anatomical resolution for deep neck spaces [4].

Chest X-ray (CXR) detects soft tissue swelling, subcutaneous air, erosion of the vertebral bodies, pneumomediastinum, displacement of the air stripe, or concurrent pneumonia, although not routinely used [5]. We found no aforementioned tissue alterations at CXR performed after surgery.

The use of magnetic resonance (MRI) is limited, although it is recommended by some authors for its higher soft tissue resolution and easier identification of multiple sites involvement in the neck than contrast tomography (CT) [6]. In DNSI, MRI shows an increased tissue contrast and an hypointense to intermediate intensity T1 signal, T2 hyperintensity, restricted diffusion, and peripheral enhancement on postcontrast T1 sequences [7].

In our patient, we performed a neck MRI as a first imaging option in order to minimize radiation exposure, better define soft tissue and deep neck space involvement, and make a prompt diagnosis of the severe complications. MRI showed a large expansive process in the retro and para-pharyngeal spaces with the involvement of the mediastinal aditus.

Clinical findings and imaging investigations are paramount to reaching a prompt diagnosis and defining the correct treatment. The mainstay of deep neck space infection or abscess management is antibiotic therapy and drainage of the abscess. Empirical parenteral antibiotics should be immediately administered and then tailored to the culture when available. Initial antibiotic therapy should include either penicillin combined with a beta-lactamase inhibitor or a beta-lactamase-resistant antibiotic (i.e., ampicillin/sulbactam or ceftriaxone). If the patient appears septic, and the treatment failure or complications occur, another antibiotic should be administered to cover MRSA (i.e., clindamycin, vancomycin, or linezolid). Parenteral antibiotics should be continued until the patient is clinically stable and afebrile for at least 24 h. After a clinical improvement of the patient, it is possible to make a shift from intravenous to oral antibiotics for 14 days (i.e., amoxicillin-clavulanate or clindamycin). When vancomycin has been added to the parenteral regimen, linezolid may be used for oral therapy [8].

In our patient, a broad-spectrum antimicrobial regimen was first administered with intravenous ceftriaxone.

The evidence suggests that DNSI in children can often be successfully managed with medical therapy alone, but life-threatening complications may occur. For this reason, incision and drainage are considered to be the gold standard for most pediatric deep-neck abscesses [9,10].

Then, the surgery was performed on our patient because of the “danger space” involvement with mediastinal extension and the high risk of developing other severe complications. According to more recent evidence [11], we decided to replace ceftriaxone with meropenem and clindamycin, providing optimal coverage for Gram-negative bacilli, anaerobes, and potentially resistant Gram-positive coccis.

If not promptly recognized, recurrent DNSI can evolve into a DNS abscess. Despite the fact that the incidence of DNSI has decreased significantly in the last decades due to the wide use of antibiotics, this condition is quite common and its correct management is still challenging. DNS abscess may lead to lethal complications such as airway obstruction, aspiration pneumonia after rupturing of the abscess, vascular involvement (thrombosis of the internal jugular vein and carotid aneurysm), pericarditis, pleural empyema, mediastinitis, and sepsis [12,13].

When recurrent episodes of inflammation of the neck occur in otherwise healthy children, anatomic anomalies should also be considered as an underlying cause.

Yang et al. [14] found that the most common cause of deep neck infection in 130 patients (44 patients with age < 18 years) was a branchial cleft abnormality.

Branchial arch anomalies are disorders of embryonic development and represent 17% of pediatric neck masses. The third and fourth branchial cleft anomalies are extremely rare, being 1–4 % of all branchial anomalies [15]. The development and differentiation of branchial apparatus occur between the third and seventh weeks of gestation.

The relationship between the involvement of mediastinum in DNSI and potential pyriform sinus fistula (PSF) depends on the embryology of the branchial arches. The third or fourth branchial fistula follows a ‘two-loop course’, originating from the caudal end of the pyriform fossa and coursing inferiorly along the tracheoesophageal groove, posterior to the thyroid gland, into the mediastinum. Hence, it loops around the aorta (if located on the left side) or the subclavian artery (if located on the right side) [16]. This anatomical connection between the pharynx and mediastinum (“danger space”) promotes the spread of the infection between these two sites.

Among several branchial cleft anomalies, the investigation of PSF in our patient was due to recurrent episodes of inflammation of the neck, thyroid gland involvement in the histological sample, and neck MRI scan. The involvement of the thyroid gland in third and fourth branchial cleft anomalies is related to the fusion of the third and fourth branchial clefts to form the pyriform sinus, an embryological event that occurs close to the thyroid gland [17].

PSF is a rare condition that commonly presents as a deep neck abscess [18]. The exact cause is still unclear and is likely related to the asymmetrical transformation of the branchial apparatus. In addition, the inflammation is commonly preceded by upper respiratory tract infections occurring repetitively in 61.1% of patients [18].

In patients with PSF, an MRI of the neck can detect a suspicious fistula tract presenting with a tunnel-like lesion or a generic thyroid gland involvement. Han et al. [19] found tunnel-like lesions between the pyriform fossa and the upper pole of the thyroid gland in 40% of patients with PSF by MRI scans; the thyroid gland was involved in 83.5% of patients.

In addition, CT of the neck could also confirm the presence and the pathway of the fistulous tract in third and fourth branchial cleft anomalies, identifying the thyroid gland involvement and excluding other neck anomalies [20]. However, its use is avoided to minimize radiation exposure.

Additionally, laryngoscopy is considered a valid diagnostic method for the diagnosis of PSF.

Bansal et al. [21] suggested that PSF could be diagnosed by laryngoscopy observing the internal opening in the pyriform fossa.

A laryngoscopy was also performed on our patient showing the presence of the internal opening of a fistulous tract in the pyriform fossa.

Compared to conventional open neck surgery, the advantages of endoscopic treatment are simpler manipulation, shorter operative time, smaller risk of surgical trauma, scar-free after operation, lower surgical cost, shorter hospital stay, and comparable efficacy. However, these endoscopic techniques greatly increase the risk of thermal damage to surrounding tissue, particularly injury of the superior laryngeal nerve and recurrent laryngeal nerve. Neck abscess incision and drainage can be performed simultaneously with this minimally invasive approach [22].

## 4. Conclusions

In a child with severe neck pain, uncomfortable conditions, and fever, deep neck space infections (DNSI) should always be included in the differential diagnosis. Children under 3 years of age are at high risk of developing DNSI complications. Indeed, restricted neck movement can be a sign of severe neck involvement. Delayed treatment is associated with significant morbidity and mortality. Neck imaging investigations, including contrast tomography (CT) or magnetic resonance (MRI), is needed for DNSI diagnosis. Some authors recommend MRI to better examine the soft tissues and identify the involvement of multiple anatomic sites with less radiant exposure than CT.

In case of DNSI complications, MRI is a useful diagnostic tool to define the extension of DNSI to the danger space and rule out the mediastinal involvement. The superior mediastinal involvement might be life-threatening. Therefore, the use of a long-lasting protocol of antibiotics would reduce the risk. Open-neck drainage surgery and antibiotic therapy led to the clinical improvement of our patient. Noteworthy, the description of this single case reduces the generalizability to the population of patients with this condition. Further studies are needed to better understand the correct management of those patients.

In clinical practice, the recurrence of DNSI should alert the physician of a potential underlying congenital abnormality. Laryngoscopy might be the best diagnostic tool to detect anatomic anomalies if imaging fails.

## Figures and Tables

**Figure 1 children-10-00017-f001:**
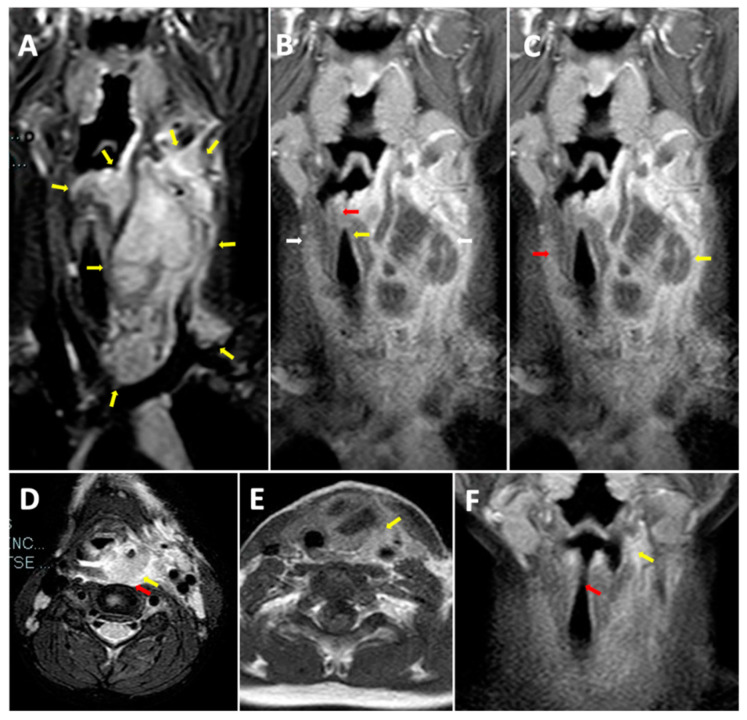
Neck MRI before treatment. (**A**) *Longitudinal STIR MRI scan of the neck*. Voluminous expansive process (32 × 25 × 85 mm Latero-lateral ×Antero posterior ×Cranio-caudal) in the retro and para-pharyngeal spaces with extension up to the mediastinal aditus. (**B**) *Longitudinal T1 MRI scan of the neck.* Compression and dislocation to the right of the larynx with marked caliber reduction (maximum diameter of 3 mm at the level of the glottic plane). (**C**) *Longitudinal T1 MRI scan of the neck.* Diffuse edema imbibition and contrast impregnation of the lateral and deep muscles and fascial planes of the bilateral cervical region (more evident on the left). (**D**) *Transverse T2 MRI scan.* Diffuse edema imbibition and contrast impregnation of the prevertebral fascia, the longus capitis muscle, and the “danger space”. (**E**) *Transverse T1 MRI scan of the neck.* Involvement of the left thyroid lobe. (**F**) *Longitudinal T1 MRI scan of the neck.* Retro and left parapharyngeal inflammatory collection with residual edema imbibition of the left lateral cervical soft tissues (maximum size of 30 × 21 × 55 mm) between the left thyroid space and the sternocleidomastoid muscle. Reduction in airway dislocation.

**Figure 2 children-10-00017-f002:**
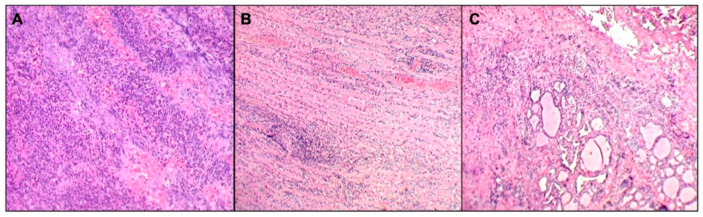
Histopathological image. Suppurative inflammatory process (**A**) involving neck striated muscle fibers, focally destroying and dissociating them (**B**), and thyroid tissue (**C**). Neither cellular atypia nor microorganisms were detected.

**Figure 3 children-10-00017-f003:**
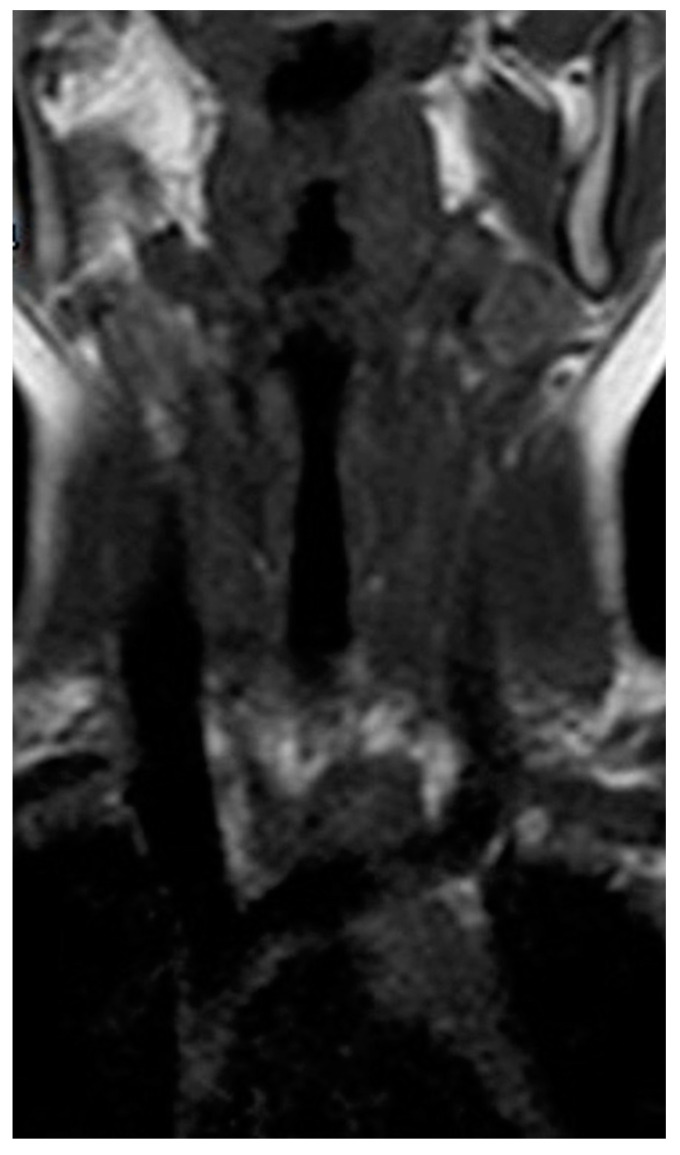
Longitudinal T1 MRI scan of the neck after treatment.

**Figure 4 children-10-00017-f004:**
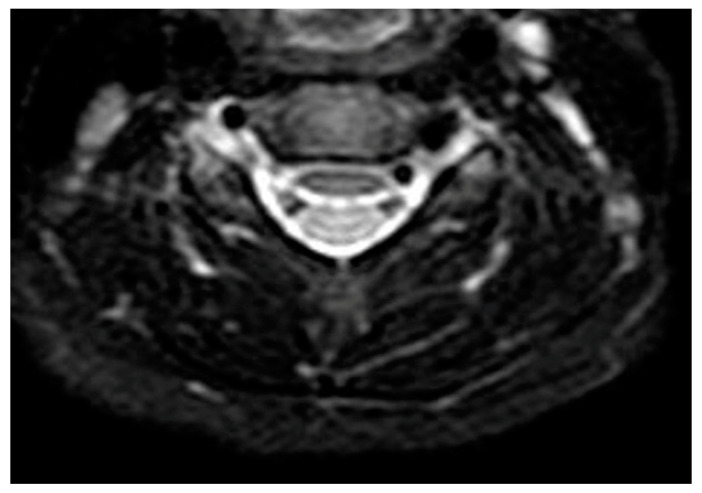
Transverse T1 MRI scan of the neck after treatment. Absence of para-laryngeal-tracheal fluid collections. Absence of para-laryngeal-tracheal fluid collections.

**Figure 5 children-10-00017-f005:**
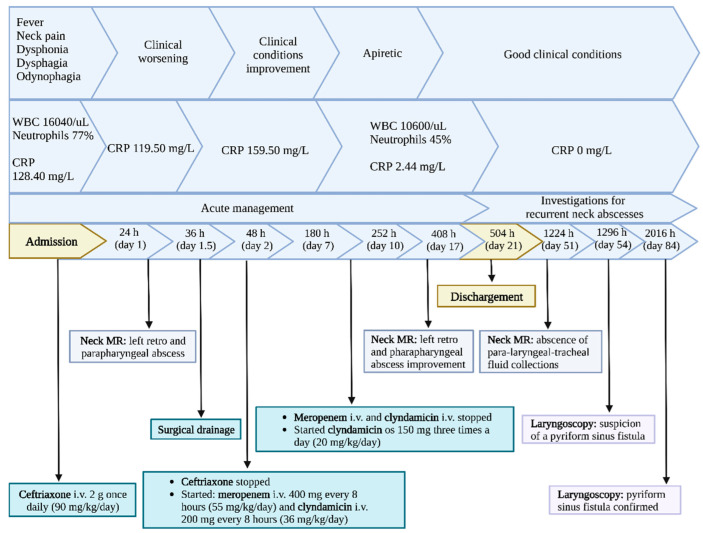
Case report timeline.

**Table 1 children-10-00017-t001:** Differential diagnosis in children presenting with neck pain and torticollis.

Disease	Diagnosis	Imaging Findings
Pharyngotonsillitis	Clinical Examination	-
Peritonsillar Abscess	Clinical Examination ± Neck Ultrasound	Complex, hypoechoic collection 5–25 mm anteromedial to the internal carotid artery
Parapharyngeal Abscess	Neck CT ± MRI	Expansive Process, Airway Compression; Dislocation; Edema, Imbibition and Contrast Impregnation of muscles/fascial planes, “Danger Space” Involvement
Retropharyngeal Abscess	Neck CT ± MRI	Expansive Process, Airway Compression; Dislocation; Edema, Imbibition and Contrast Impregnation of muscles/fascial planes, “Danger Space” Involvement
Acute Epiglottitis	Clinical Examination ± Neck CT	Marked edema and thickening of the epiglottis and aryepiglottic folds with narrowing of the airway
Bacterial Tracheitis/Croup	Clinical Examination	Normal
CNS Tumor	Brain CT ± MRI	Expansive Lesions, Cyst/Nodule, Calcifications, Necrosis, Hemorrhage
CNS Infection	Brain CT ± MRI	Ring Enhancing Lesions, Basal Ganglia Alterations, White/Gray Matter T2 Hyperintensity
Apical Pneumonia	Clinical Examination ± CXR	Upper Lobes Opacity
Fractures/Dislocations	Cervical X-ray (AP + LL Projection)	Irregular and Non-corticated Vertebral Line Vertebral Disk Space Widening Bilateral Interfacetal Dislocation
Neck Muscle Hematomas	Neck MRI	Soft Tissue/Muscle Swelling
Cervical Dystonia	Clinical Examination ± EMG ± MRI	Possible Structural Brain Alterations
Foreign Body Ingestion	CXR + Abdomen X-ray (if Radiopaque)	Radiopaque Foreign Body within Gastro-Intestinal Tract
Acute Esophagitis	History + Clinical Examination ± Endoscopy ± Esophageal pH Monitoring	-
Malignant Etiology	CT/MRI + Biopsy Confirmation	Expansive Lesions with Earlier and Faster Uptake of Contrast Material
Ocular Dysfunction	Eye Examination + Vision Testing	-
Temporomandibular Arthritis in Juvenile Idiopathic Arthritis	Clinical Examination	-
Kawasaki Disease	Clinical Examination	-
Autoimmune Granulomatous Diseases	Clinical Examination + Laboratory studies	-
Thyroiditis	Clinical Examination + Laboratory studies + Thyroid US	Nodular lesions, nonnodular lesions
Pediatric Thyroid Cancer	History + Clinical Examination + Laboratory studies + Thyroid US + Neck CT +Fine Needle Aspiration	Solid nodules, cystic lesions, lesions with a thick irregular halo
Pneumomediastinum	Clinical Examination + CXR	Thymic sail sign, “ring around the artery” sign, tubular artery sign, double bronchial wall sign, extrapleural sign
Child abuse	History + Clinical Examination ± CXR ± Neck US	Bone or soft tissue lesions

**Table 2 children-10-00017-t002:** Acute management additional exams and investigations for recurrent neck abscesses.

Acute Management Additional Exams	Results
Serology for Epstein-Barr virus	Negative
Serology for Cytomegalovirus	Negative
Serology for *Toxoplasma gondii*	Negative
Serology for *Bartonella henselae*	Negative
Serology for Adenovirus	Negative
Anti-streptolysin O	Negative
Pharyngeal swab for *Group A Beta-Hemolitic Streptococcus*	Negative
Culture for *Staphylococcus Aureus*	Negative
Culture for *Streptococcus viridans*	Negative
Culture for *Escherichia coli*	Negative
Culture for *Klebsiella pneumoniae*	Negative
Culture for *Veilonella*	Negative
Culture for *Haemofilus influenzae*	Negative
Culture for *Enterobacter*	Negative
Blood culture	Negative
Interferon gamma release assay (IGRA test)	Negative
Peripheral venous blood smear	Negative for blood and blood-related diseases
**Investigations for recurrent neck abscesses**	**Results**
Serum immunoglobulin level (IgM, IgA, IgG)	Normal level
B cell phenotyping profile	No humoral and B-cell immunity abnormality
T cell surface markers by flow cytometry	No cellular immunodeficiency

## Data Availability

Not applicable.

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
