# Peer review of "A 5-Year-Old Child with a Deep Neck Abscess Complicated by Laryngeal Obstruction"

_children, 2022, doi:10.3390/children10010017_

Round 1
Reviewer 1 Report
Dear authors,
Thanks for this manuscript in which you describe the case of a 5 y/o girl with recurrent neck abscesses. It is certainly a very interesting case, but a first comment is that the text is very (++) lengthy; I would therefore consider writing it more condensed (I give some specific recommendations, but please critically revise the entire text). Furthermore, there are 2 different aspects to the case that you describe in one. The presentation would be more structured if it was presented in 2 parts (e.g. 'acute management' vs. 'investigations for recurrent neck abscesses). Last, the antibiotic management in this case does not appear to be rational. Certainly, there was a patient who was not improving, therefore the choice to escalate could be logical. However, the different other pieces (recurrences, imaging findings) strongly point towards a source control issue rather than an insufficient spectrum. This should clearly be described.
Specific comments:
Page 2: "Upon admission .... submandibular lymphadenopathy." The phrase "poor general condition" is rather unspecific. Probably the term "uncomfortable" would be better here.
Page 2: Think that the differential of neck pain with fever should also include non-infectious diseases (e.g. Kawasaki).
Page 3: "Serum immunoglobulin ... which was normal." I am not sure what this adds to the evaluation of an acute infection; this is more something to describe for the work up of recurrent infections at the same location.
Page 3: "We decided ... malignant cancer" this is a very lengthy text that could perfectly be replaced by a table.
Page 3: "Cultural exams" should be "all cultures" (similar to page 8 "cultural results" that should be "culture")
- Page 8: "in combination with a drug highly effective against anaerobes" Most beta lactamase inhibitors have good coverage against anaerobes; therefore the combination of ampi/sulbactam + metronidazole would be redundant.
- Names of bacteria should be in italic.
Author Response
- “The text is very lengthy; consider writing it more condensed”.
We thank the referee for the comment, the text is now more condensed.
- “The presentation would be more structured if it was presented in 2 parts: acute management and investigations for recurrent neck abscesses”.
We thank the referee for the comment. The manuscript has been presented as indicated.
- “The antibiotic management does not appear to be rational; Certainly, there was a patient who was not improving, therefore the choice to escalate could be logical; however, the different other pieces (recurrences, imaging findings) strongly point towards a source control issue rather than an insufficient spectrum; this should clearly be described”.
We thank the referee for the comment. We decided to use the described antibiotic management according to the most recent indications in the literature (Ref. 13 Jain H. et al. Retropharyngeal Abscess. [Updated 2022 Sep 12]. In: StatPearls) and our choice was guided by the clinical response of the patient, mostly after the surgery. We better explained that in the manuscript (line 284 in the text).
- Page 2: “Upon admission…submandibular lymphadenopathy”. The phrase “poor general condition” is rather unspecific; probably the term “uncomfortable” would be better here.
We thank the referee for the comment. The phrase “poor general condition” has been replaced with “uncomfortable”.
- Page 2: “think that the differential of neck pain with fever should also include non-infectious diseases”.
We thank the referee for the comment. Other non-infectious causes of neck pain and torticollis in children has been included (Kawasaki Disease, temporomandibular arthritis in Juvenile Idiopathic Arthritis, Autoimmune Granulomatous Disease, Thyroiditis, Pediatric Thyroid Cancers, Pneumomediastinum, Child abuse).
- Page 3: "Serum immunoglobulin ... which was normal." I am not sure what this adds to the evaluation of an acute infection; this is more something to describe for the work up of recurrent infections at the same location.
We thank the referee for the comment. Serum immunoglobulin level, B cell phenotyping profile, and T cell surface markers have been reported in “investigations for recurrent neck abscesses” section.
- Page 3: "We decided ... malignant cancer" this is a very lengthy text that could perfectly be replaced by a table.
We thank the referee for the comment. This text has been removed and summarized in table 2.
- Page 3: "Cultural exams" should be "all cultures" (similar to page 8 "cultural results" that should be "culture").
We thank the referee for the comment. “Cultural exams” has been changed in “all cultures” and “cultural results” has been changed in “culture”.
- Page 8: "in combination with a drug highly effective against anaerobes" Most beta lactamase inhibitors have good coverage against anaerobes; therefore the combination of ampi/sulbactam + metronidazole would be redundant.
We thank the referee and followed his/her important suggestion. The text “in combination with a drug highly effective against anaerobes” has been removed.
- Names of bacteria should be in italic.
We thank the referee and followed his/her important suggestion. Names of bacteria has been changed to italic.
Reviewer 2 Report
Manuscript Number: children-2060203-peer-review-v1
Title:
A 5-year-old child with fever and neck pain
1. Yes, this subject is useful for publication in Children – the main topic is
Case report od deep neck infection.
2. Author demonstrated
clinical case of a retro and para-pharyngeal abscess in a healthy 5-year-old child complicating by compression and dislocation of the larynx.
3. The design and results are clearly presented. Introduction is missing.
Table 1 is not correct Peritonsillar, para and retropharyngeal abscesses are missing. Lateral neck Xray of acute epiglottitis is not recommended nowadays.
4. Discussion is logical and correct.
6. References are current and pertinent.
7. Figures are in good quality. Figure of laryngoscopy showed the suspicion of a pyriform sinus fistula
Author Response
- Yes, this subject is useful for publication in Children – the main topic is Case report of deep neck infection.
We thank the referee for his/her important opinion.
- Author demonstrated clinical case of a retro and para-pharyngeal abscess in a healthy 5-year-old child complicating by compression and dislocation of the larynx.
We thank the referee for his/her important opinion.
- The design and results are clearly presented. Introduction is missing.
We thank the referee and followed his/her important suggestion. Introduction has been inserted
- Table 1 is not correct Peritonsillar, para and retropharyngeal abscesses are missing. Lateral neck Xray of acute epiglottitis is not recommended nowadays.
We thank the referee and followed his/her important suggestion. Peritonsillar, para and retropharyngeal abscess have been entered in the table. Lateral neck X-ray of acute epiglottitis has been removed.
- Discussion is logical and correct.
We thank the referee for his/her important opinion.
- References are current and pertinent.
We thank the referee for his/her important opinion.
- Figures are in good quality. Figure of laryngoscopy showed the suspicion of a pyriform sinus fistula
We thank the referee for the comment. Unfortunately, we didn’t have the figure of laryngoscopy.
Reviewer 3 Report
1- last line of the abstract: replace “whether” with “when”.
2- The authors should clarify whether the child had obvious neck swelling, obvious limitation of neck movements and abnormal head posture or not in the symptoms section.
3- was there neck rigidity on exam?
4- the authors excluded malignant cancer with negative peripheral venous blood smear. How is that possible?
5- what is meant by “the long muscle of the head”?
6- how was the patient intubated for surgery? Trans oral or trans nasal?
7- there is a paragraph that is repeated “six day, after the abscess … follow up program”.
8- minor english language edits are required throughout the manuscript.
9- Figure 3 has two titles. The authors should choose only one.
10- there is a figure “the case timeline” without any title.
Author Response
- Last line of the abstract: replace “whether” with “when”.
We thank the referee for the suggestion. The word “whether” was replaced with “when”.
- and 3. The authors should clarify whether the child had obvious neck swelling, obvious limitation of neck movements and abnormal head posture or not in the symptoms section. Was there neck rigidity on exam?
We thank the referee for the comment. The presence of the suggested symptoms and signs has been clarified in the manuscript.
- The authors excluded malignant cancer with negative peripheral venous blood smear. How is that possible?
We thank the referee for the comment. We are sorry for the misunderstanding. Peripheral venous blood smear was negative for blood and blood related diseases. Malignant cancer cancer was excluded by histological examination and microscopic exam of surgical samples.
- What is meant by “the long muscle of the head”?
We thank the referee for the comment. The long muscle of the head is the longus capitis muscle. The term “long muscle of the head” has been changed to “longus capitis muscle” in the text for a more accurate definition.
- How was the patient intubated for surgery? Trans oral or trans nasal?
The patient underwent oral intubation without complications, despite the high risk of abscess rupture in these patients.
- There is a paragraph that is repeated “six day, after the abscess … follow up program”.
We thank the referee for the comment. The repeated paragraph has been removed.
- Minor english language edits are required throughout the manuscript.
We thank the referee for the comment. Minor english language edits have been applied throughout the manuscript.
- Figure 3 has two titles. The authors should choose only one.
We thank the referee for the comment. Only one title for figure 3 has been maintained.
- There is a figure “the case timeline” without any title.
We thank the referee for the comment. The title for the case timeline has been added.
Round 2
Reviewer 1 Report
Dear authors,
Thanks for this thorough revision of your interesting case report. The quality has definitely been improved. However, albeit shortened compared to the previous version- it is still too long. Furthermore, I would respectfully recommend you to use an English editing service to enhance the readability of the manuscript. I will outline some suggestions below- but these are by no means exhaustive.
- Introduction: thanks for condensing the text. However, it is still too long and it contains a lot of information that is in the case presentation. Therefore, please revise the introduction section ahead; it is totally acceptable if it were just a couple of sentences.
- Abstract: "in a 5 year old" should be followed by ", complicated by"
- Case report: Consider rephrasing "Acute management" by "Acute presentation".
- Lines: "Regarding the differential .... swelling and mass (table 1)"; please omit this paragraph- it is simply re-iterating table 1.
- Page 4: "Therefore, we performed .... imaging tests." Consider rewriting as: "Therefore, we performed neck and brain MRI"
- Page 4: "During the hospitalization ... which were normal." I assume that the child did not perform the echo but rather underwent.
- Discussion: "Our patient complained with ... dysphonia as atypical." should be "Our patient presented with"
- MRSA: please consistently remove capital in Aureus (as it should be written as aureus)
- Viruses should NOT be written in Italics
- Table 2: please report IGRA as negative and omit "for TB disease)
Author Response
The quality has definitely been improved. However, albeit shortened compared to the previous version- it is still too long. Furthermore, I would respectfully recommend you to use an English editing service to enhance the readability of the manuscript.
We thank the referee for the comment. The manuscript has been presented as indicated and has been. A native speaker reviewed and revised the article.
- Introduction: thanks for condensing the text. However, it is still too long and it contains a lot of information that is in the case presentation. Therefore, please revise the introduction section ahead; it is totally acceptable if it were just a couple of sentences.
We thank the referee for the comment. We reviewed the introduction as indicated.
- Abstract: "in a 5 year old" should be followed by ", complicated by"
We thank the referee for the comment. “Complicated by” has been added after “in a 5 year old”.
- Case report: Consider rephrasing "Acute management" by "Acute presentation".
We thank the referee for his/her suggestion. “Acute management” has been changed with “Acute presentation”.
- Lines: "Regarding the differential .... swelling and mass (table 1)"; please omit this paragraph- it is simply re-iterating table 1.
We thank the referee for the comment. “Regarding the differential…swelling and mass (table 1)” paragraph has been removed.
- Page 4: "Therefore, we performed .... imaging tests." Consider rewriting as: "Therefore, we performed neck and brain MRI"
We thank the referee for the suggestion. “Therefore, we performed…imaging tests” has been changed with “Therefore, we performed neck and brain MRI”.
- Page 4: "During the hospitalization ... which were normal." I assume that the child did not perform the echo but rather underwent.
We thank the referee for the comment. “Perform” has been changed with “underwent”.
- Discussion: "Our patient complained with ... dysphonia as atypical." should be "Our patient presented with"
We thank the referee for the comment. “Complained” has been changed with “presented”.
- MRSA: please consistently remove capital in Aureus (as it should be written as aureus)
We thank the referee for the comment. Capital in “Aureus” has been removed and has been written as “aureus”.
- Viruses should NOT be written in Italics
We thank the referee for the comment. Viruses have been written as indicated.
- Table 2: please report IGRA as negative and omit "for TB disease)
We thank the referee for the comment. “Negative for TB disease” has been changed with “Negative”.